# The Role of Lipoprotein(a) in Peripheral Artery Disease

**DOI:** 10.3390/biomedicines12061229

**Published:** 2024-06-01

**Authors:** Nicholas Pavlatos, Dinesh K. Kalra

**Affiliations:** 1Department of Internal Medicine, University of Louisville School of Medicine, Louisville, KY 40202, USA; nicholas.pavlatos@louisville.edu; 2Division of Cardiology, University of Louisville School of Medicine, Louisville, KY 40202, USA

**Keywords:** peripheral artery disease, lipoprotein(a), atherosclerosis, lipids

## Abstract

Lipoprotein(a) is a low-density-lipoprotein-like particle that consists of apolipoprotein(a) bound to apolipoprotein(b). It has emerged as an established causal risk factor for atherosclerotic cardiovascular disease, stroke, and aortic valve stenosis through multifactorial pathogenic mechanisms that include inflammation, atherogenesis, and thrombosis. Despite an estimated 20% of the global population having elevated lipoprotein(a) levels, testing remains underutilized due to poor awareness and a historical lack of effective and safe therapies. Although lipoprotein(a) has a strong association with coronary artery disease and cerebrovascular disease, its relationship with peripheral artery disease is less well established. In this article, we review the epidemiology, biology, and pathogenesis of lipoprotein(a) as it relates to peripheral artery disease. We also discuss emerging treatment options to help mitigate major adverse cardiac and limb events in this population.

## 1. Introduction

Lipoprotein(a) [Lp(a)] is a lipoprotein structurally similar to low-density lipoprotein and is characterized by apolipoprotein(a) [apo(a)] covalently bound to apolipoproteinB-100 (apoB-100). It was first described by Norwegian geneticist Kare Berg and his colleagues in 1963 [1]. He was able to recognize that Lp(a) levels were genetically determined and, by 1974, had suggested that Lp(a) contributes to coronary artery disease (CAD) [2]. Fifty years later, it has been established that Lp(a) is an independent causal risk factor for CAD, cerebrovascular disease (CeVD), and aortic stenosis [3]. The National Lipid Association pooled data from large prospective, population-based studies and found that when compared to patients with Lp(a) < 5 mg/dL, those with Lp(a) Lp(a) ≥ 120 mg/dL were at a 5-fold risk of coronary artery stenosis, 3- to 4-fold risk of myocardial infarction, 1.7-fold risk of carotid stenosis, 1.6-fold risk of ischemic stroke, 3-fold risk of aortic stenosis, and 1.6-fold risk of femoral artery stenosis, suggesting CAD likely has a greater association with Lp(a) than PAD and CeVD [4]. Lp(a) levels are 90% genetically determined and not appreciably affected by diet or exercise; therefore, serum levels typically remain stable in individuals over their lifetime [5,6]. Although the consequences of elevated Lp(a) levels are well described, testing remains sparse. Additionally, treatment options are limited as traditional therapies for dyslipidemias appear to be ineffective in the management of elevated Lp(a) levels [7].

Peripheral artery disease (PAD) is a progressive atherosclerotic disorder characterized by narrowing of the arteries other than those that supply the heart or brain, typically affecting the lower extremities [8]. It is estimated that over 200 million people suffer from PAD worldwide, with the majority of patients being asymptomatic [9]. Common symptoms include exertional leg pain, discomfort, and cramping known as claudication but also include more atypical symptoms such as pain at rest. Complications of PAD can be devastating and include infection, critical limb ischemia (CLI), acute limb ischemia (ALI), amputation, and death [10]. Studies have shown that PAD significantly increases mortality with a 10-year all-cause mortality of 27% for the reference range, 56% for asymptomatic PAD, 63% for intermittent claudication, and 75% for limb ischemia [11]. Additionally, PAD imposes a substantial burden on healthcare spending with an average annual expenditure per PAD patient of USD 11,533 compared to USD 4219 in patients without PAD [12]. Despite data implicating elevated serum Lp(a) levels and the development of PAD, this association remains underrecognized [13]. Major guidelines are lacking in both the diagnosis and management of elevated Lp(a) levels in the setting of PAD.

## 2. Epidemiology

According to the National Heart Lung and Blood Institute report published in 2018, an estimated 1.43 billion people have Lp(a) levels ≥ 50 mg/dL. Although data are lacking in regard to the prevalence of elevated Lp(a) levels in PAD, the Copenhagen General Population study was able to show that elevated Lp(a) levels results in a stepwise increase in PAD. They found that in smokers aged 70–79 years with Lp(a) < 50th and ≥99th percentile, the 10-year risk for PAD among females was 8% and 21%, while among males, it was 11% and 29% [14]. The prevalence of elevated Lp(a) varies between continents due to LPA gene remodeling and different migration patterns of people with different sized isoforms of Lp(a) [15]. Studies have shown that South Asians and Blacks of sub-Saharan African descent have higher levels of Lp(a) when compared to Hispanics, East Asians, and Whites [16,17,18]. The Multi-Ethnic Study of Atherosclerosis (MESA) cohort investigated four ethnic groups and reported median Lp(a) levels among Blacks were 35.1 mg/dL, Whites 12.9 mg/dL, Hispanics 13.1 mg/dL, and Chinese 12.9 mg/dL [19]. Although they did not study South Asians, the Mediators of Atherosclerosis in South Asians Living in America (MASALA) cohort reported a median Lp(a) level of 17.0 mg/dL [20]. Neither study reported the prevalence of PAD in their study populations. Despite Lp(a) being a known independent risk factor for cardiovascular disease and aortic stenosis, testing persists at low rates even in those diagnosed with such conditions. An observational study that reviewed over 5 million patients from 2012 to 2021 at the University of California Health System Warehouse found that only 0.3% of those patients had Lp(a) testing. Additionally, the rate of Lp(a) testing in those with a known diagnosis of PAD was 2.0%. Patients with a history of ischemic heart disease, aortic stenosis, and stroke fared no better as they were tested at similar rates of 2.9%, 3.1%, and 1.8%, respectively [21]. Another observational study analyzing data from 2015 to 2019 of healthcare systems participating in the National Patient-Centered Clinical Research Network found that only 0.06% of patients per year were tested for Lp(a). They also found that out of those diagnosed with atherosclerotic cardiovascular disease, testing was less common in those with PAD compared to ischemic stroke or myocardial infarction (MI) [22]. Poor utilization of Lp(a) testing in the clinical setting is likely a result of low provider awareness, lack of recommendations in major guidelines, limited evidence of benefit in lowering Lp(a), and no currently available approved drugs for this indication.

## 3. Structure/Genetics

Lp(a) is a low-density-lipoprotein-like molecule made up of two protein-containing moieties, apo(a) and apoB-100, covalently bound by a disulfide bridge (Figure 1) [23]. ApoB-100 is synthesized in the liver and is a key structural component of atherogenic lipoproteins such as low-density lipoprotein, very-low-density lipoprotein, and intermediate-density lipoprotein [24]. The apo(a) chain is structurally similar to plasminogen and is believed to promote inflammation, endothelial dysfunction, and calcification by attracting oxidized phospholipids [25]. Notable components of apo(a) are kringle domains which are triple-looped structures stabilized by three disulfide linkages. These are also present in other coagulation factors, most notably, plasminogen, prothrombin, and urokinase [23,26]. Each apo(a) contains one copy of kringle IV type 1, kringle IV types 3–10, and kringle V domains, and multiple copies of kringle IV type 2 domains, ranging from 1 to over 40 copies. This heterogeneity in the number of kringle IV type 2 repeats makes apo(a) highly polymorphic in size. This is significant as larger apo(a) isoforms are less efficiently secreted by hepatocytes, resulting in an inverse relationship between apo(a) isoform size and Lp(a) levels [27]. Plasma Lp(a) levels are highly variable within the same population, ranging from less than 0.1 mg/dL to over 300 mg/dL with up to 80% of variability being attributed to the apo(a) size polymorphisms [28]. The major gene locus for determining apo(a) structure, and therefore Lp(a) concentrations, is LPA which is located on chromosome 6q26-q27 [29]. Genome-wide association studies have identified single-nucleotide polymorphisms at the LPA locus (rs10455872 and rs7452960) that have been proven to increase both serum Lp(a) levels and the development of PAD [30,31,32].

## 4. Pathogenesis

High levels of Lp(a) are thought to promote atherosclerosis through at least five pathways: (1) attracting leukocytes to the vascular endothelium by increasing the expression of adhesion molecules on vascular endothelial cells and inducing the production of chemotactic agents by them, (2) infiltration of vessel wall and foam cell formation, (3) vascular smooth muscle cell proliferation, (4) endothelial cell dysfunction, and (5) plaque inflammation (Figure 2).

Lp(a) increases the expression of multiple adhesion molecules such as vascular cell adhesion molecule, intercellular cell adhesion molecule, and E-selectin. These surface adhesion molecules allow for corresponding receptors on leukocytes to bind and migrate into the arterial intima [33]. Additionally, Lp(a) induces production of monocyte chemotactic protein in vascular endothelial cells which attracts monocytes and other cytokines, all of which contribute to atherosclerosis [34].

Lp(a) has been detected in the intima and subintima of blood vessels [35]. The proposed mechanism by which this occurs is through the lysine binding site of kringle IV of apo(a) which allows Lp(a) to anchor to the extracellular matrix within the arterial wall [36]. Once inside the arterial wall, Lp(a) undergoes oxidative, lipolytic, and proteolytic changes before uptake by monocytes to form foam cells [37]. Over time, the foam cell’s ability to manage extra lipoproteins decreases, resulting in endoplasmic reticulum stress and reactive oxygen species. Eventually, an apoptotic cascade ensues in which foam cells release prothrombotic molecules, cellular proteases, inflammatory cytokines, and oxidized LDL free radicals, all of which act as irritants leading to plaque formation [38].

Lp(a) promotes vascular smooth muscle proliferation by blocking the conversion of plasminogen to plasmin, thereby preventing plasmin-mediated activation of Transforming Growth Factor Beta, an inhibitor of smooth muscle migration [39].

Elevated levels of Lp(a) inhibit nitric oxide synthesis and thereby promote endothelial cell dysfunction. Nitric oxide has many anti-atherogenic characteristics such as inhibition of platelet activation, smooth muscle proliferation, and reduction in endothelial permeability [37]. Additionally, elevated Lp(a) levels impair endothelium-dependent vasodilation. This phenomenon has been observed even in the absence of atherosclerosis, indicating a direct inhibitory effect or mechanism at play by which Lp(a) increases vascular permeability and allows direct access of acetylcholine to vascular smooth muscle cells [40].

Elevated levels of Lp(a) also results in plaque inflammation and instability by mediating macrophages to release interleukin-8. Interleukin-8 is an inflammatory cytokine within atherosclerotic plaque and has chemotactic activity towards smooth muscle cells, monocytes, T-cells, and neutrophils [41]. Additionally, interleukin-8 decreases macrophage expression of metalloproteinases which cleave apo(a). Disinhibition of these metalloproteinases leads to plaque inflammation and eventual rupture [42].

Beyond atherosclerosis, Lp(a) is thought to promote thrombosis via the inhibition of plasmin generation, inhibition of Tissue Factor Pathway Inhibitor (TFPI), and increased expression of Plasminogen Activator Inhibitor (Figure 3). Due to apo(a)’s structural similarity to plasminogen, it is theorized that Lp(a) may interfere with fibrinolysis by competitive inhibition of plasmin generation on endothelial cells, monocytes, and platelets. This characteristic of Lp(a) is not well established and may have little clinical relevance [43]. Additionally, Lp(a) inhibits TFP1, an anticoagulant protein that works by inhibiting early phases of procoagulant responses. TFP1 is present on platelets, monocytes, vascular smooth muscle cells, endothelial plaques, and endothelial cells. Lp(a) has been shown to bind and inactivate TFPI, leading to unopposed tissue factor mediated thrombosis [44]. Lastly, Lp(a) causes increased expression of Plasminogen Activator Inhibitor at the endothelial cell surface, thereby inhibiting fibrinolysis [45].

## 5. Diagnostic Testing

The measurement of Lp(a) is challenging in that there is a large variation in not only lipid composition of Lp(a) but also in the size of the Apo(a) moiety between individuals. Lp(a) can be measured in molar concentration (nmol/L), mass units (mg/dL), or the seldomly used Lp(a) cholesterol mass [46,47]. Molar concentration by enzyme-linked immunosorbent assay remains the gold standard and is a better predictor of ASCVD than mass units. Although conversions exist between the two units of measure, these are inaccurate and should be avoided [48]. The European Atherosclerotic Society consensus statement reports that <75 nmol/L or <30 mg/dL is considered normal, 50–125 nmol/L or 30–50 mg/dL is intermediate, and >125 nmol/L or >50 mg/dL is abnormal [49]. The ACC/AHA agree that Lp(a) ≥ 125 nmol/L or ≥50 mg/dL is considered abnormal and is a risk factor for ASCVD [50].

Serum Lp(a) levels are genetically determined and remain stable over time; therefore, there is little utility in re-checking Lp(a) unless a provider wishes to evaluate the efficacy of treatment. Lp(a) should be checked at steady states as it can be altered by multiple non-genetic factors. Levels are increased during chronic inflammatory conditions, pregnancy, hypothyroidism, kidney disease, and growth hormone therapy. Levels are decreased in severe acute inflammatory conditions, hyperthyroidism, and liver disease [51].

## 6. Screening

There is a large amount of evidence showing that the majority of PAD remains undetected in clinical practice; therefore, there has been interest in detecting PAD through routine screening. Different guidelines have variable recommendations regarding screening for PAD. In 2018, the United States Preventive Services Task Force concluded that there was insufficient evidence to screen asymptomatic patients for PAD [52]. Although the ESC/ESVS guidelines make no comment on screening of PAD in asymptomatic individuals, they recommend against screening patients with known atherosclerotic disease of other vascular beds as this would likely not change management strategy [53]. In a recent AHA statement, it is recommended to screen for PAD with ABI in high-risk adults such as those ≥ 65 years of age and 50–64 years of age with traditional risk factors. Traditional risk factors include smoking, diabetes, hypertension, and dyslipidemia [10]. No major guidelines comment on screening for PAD in the setting of elevated Lp(a) levels.

Given its role as an independent risk factor for ASCVD, many guidelines have published their recommendation on Lp(a) screening. The AHA/ACC state there is a relative indication for checking Lp(a) levels in those with family or personal history of ASCVD that is not explained by another risk factor and in those aged 40–75 without diabetes mellitus who have intermediate risk of ASCVD [50]. The ESC/EAS believe it may be helpful to check Lp(a) once in adults in order to identify those with very high inherited levels of Lp(a). They also state that it is reasonable to check Lp(a) in those who have family history of premature ASCVD and to reclassify on the border of risk categories in order to determine treatment strategies [54]. Other societies such as the National Lipid Association and Canadian Cardiovascular Society take a more aggressive approach in screening, recommending that everyone should obtain an Lp(a) measurement as part of initial lipid screening [55,56]. Just as no major guidelines recommend screening for PAD in the setting of elevated Lp(a) levels, there are conversely no recommendations for screening Lp(a) in the setting of PAD.

## 7. Lp(a) in PAD

Perhaps the largest study assessing the role of Lp(a) in PAD used samples from the Copenhagen General Population Study, a prospective cohort study in which 108,446 patients aged 20 to 100 years were enrolled between 2003 and 2015. Of the 70,317 individuals who had a measurement of plasma Lp(a), 1162 developed PAD, and 159 had lower-extremity amputation. Individuals with Lp(a) ≥ 90th percentile versus <50th percentile were significantly more likely to be diagnosed with PAD in the hospital (HR 2.99, 95% CI 2.09–4.30), experience major adverse limb events (MALE) (2.35, 95% CI 1.04–5.29), and undergo lower-extremity amputation (HR 1.62, 95% CI 1.00–2.62). For the purpose of this study, MALE was defined as revascularization and/or limb amputation. The study also looked into the polymorphisms in Lp(a) structure and found that those with kringle IV type 2 repeats ≤ 5th percentile, and thus higher genetically determined Lp(a) levels, were more likely to develop PAD than those with kringle IV type 2 repeats > 50th percentile (1.39, 95% CI 1.17–1.64) [14]. The effects of Lp(a) on MALE were further corroborated by a single-center, retrospective, cohort study consisting of 16,153 hospitalized patients in France between 2000 and 2020. Serum Lp(a) levels were stratified as normal (<50 mg/dL), high (≥50 mg/dL but <134 mg/dL), and very high (≥134 mg/dL). They found not only a statistically significant stepwise increase in prevalence of PAD between the three groups but also a 5-year cumulative incidence of MALE of 5.44% (95% CI, 3.79%–7.07%) in the very high group, 4.43% (95% CI, 3.68%–5.16%) in the high group, and 3.33% (95% CI, 2.96%–3.70%) in the normal group [57]. These studies indicate that not only is Lp(a) associated with an increase in PAD but also an increase in MALE.

The European Prospective Investigation of Cancer (EPIC)-Norfolk obtained Lp(a) levels of 18,720 participants between the ages of 39 and 79. Of these participants, 596 were diagnosed with PAD. They analyzed Lp(a) logarithmically to examine it as a continuous variable and found the hazard ratio for PAD for a 2.7-fold increase in Lp(a) (equating for 1 standard deviation in Lp[a]) was 1.37 (95% CI 1.25–1.50) [58].

A case–control study which utilized the Linz Peripheral Arterial Disease (LIPAD) data investigated both serum Lp(a) levels and low molecular weight phenotypes of apo(a) in PAD. They measured serum Lp(a) levels and apo(a) phenotypes in 213 patients with symptomatic PAD and compared them to controls matched for age, sex, and presence of diabetes. Low molecular weight phenotypes were defined as apo(a) isoforms with 11–22 kringle IV type 2 repeats, whereas high molecular weight apo(a) isoforms had ≥22 kringle IV type 2 repeats. It was found that those with Lp(a) ≥ 19.5 g/dL were significantly more likely to have PAD (OR 3.73, 95% CI 2.08–6.67). Furthermore, low molecular weight phenotypes of apo(a) were more likely to have PAD (OR 2.21, 95% CI 1.33–3.67) [59].

An Australian study which looked at 1472 people with PAD in the outpatient setting found that patients with Lp(a) ≥ 30 mg/dL were more likely to undergo lower limb revascularization alone (HR 1.33, 95% CI 1.06–1.66) as well as any PAD operation (HR 1.20, 95% CI 1.02–1.41). They did not find an increase in all-cause mortality or major adverse cardiac events (MACEs). Lp(a) levels ≥ 50 mg/dL were also associated with increased risk of lower limb revascularization (HR 1.29, 95% CI 1.00–1.65) but had no association with mortality, MACE, or any PAD operations [60].

The Edinburgh Artery Study evaluated the development of PAD in the general population, taking into account various blood markers including Lp(a). Lp(a) showed a strong independent association in the development of PAD. In order to compare between different hazard ratios, Lp(a) was divided by the inter-tertile range. After adjusting for cardiovascular risk factors and baseline atherosclerotic cardiovascular disease, elevated Lp(a) levels were associated with the development of PAD with a hazard ratio corresponding to an increase equal to the inter-tertile range of 1.22 [61].

The InCHIANTI study studied the association of elevated Lp(a) levels in prevalent and incident cases of PAD in 1002 Italian subjects aged 60–96. After adjusting for confounding variables, they found that participants in the highest quartile of Lp(a) distribution were more likely to have PAD (OR: 1.38, 95% CI 1.01–3.33). This association significantly increased if the diagnostic criteria of PAD was strengthened from ABI < 0.90 to <0.70 (OR: 3.80, 95% CI 1.50–9.61) [62].

FRENA (Factores de Riesgo y Enfermedad Arterial) is a Spanish registry of outpatients with CAD, CeVD, or PAD. Of the 1503 patients recruited, 814 (54%) had Lp(a) levels < 30 mg/dL, 319 (21%) had 30–50 mg/dL, and 370 (25%) had ≥50 mg/dL. It was found that patients with Lp(a) between 30 and 50 mg/dL were at higher risk of limb amputation (HR: 3.18, 95% CI 1.36–7.44), while those with levels ≥ 50 mg/dL were at even higher risk (HR: 22.7, 95% CI 9.38–54.9), suggesting higher Lp(a) levels result in a stepwise increase in PAD [63].

In a retrospective study looking at the association between Lp(a) and PAD in patients who had undergone coronary artery bypass graft and had no history of diabetes were divided into two groups: Lp(a) < 30 mg/dL and Lp(a) ≥ 30 mg/dL. They found that females with Lp(a) > 30 mg/dL had a significant higher risk of developing PAD than those with Lp(a) < 30 mg/dL (OR: 2.589, 95% CI 1.376–4.870). No such association was found in male patients (OR: 0.965, 95% CI 0.628–1.364). Although this study had a limited sample size, it was the first to suggest a gender deviation in the association between Lp(a) and PAD [64].

In the largest genome-wide association study of PAD which investigated 19 genetic loci in 31,307 individuals with PAD and 211,753 controls, 19 PAD loci were identified. Of the 19 loci, the LPA gene was found to be the most associated with PAD (OR 1.26, 95% CI 1,22–1.30) [32].

Although the majority of the data point toward an association between elevated Lp(a) levels and PAD, not all studies have been able to prove this. In a prospective cohort study assessing 14,916 healthy male physicians aged 40 to 84, novel risk factors for systemic atherosclerosis were obtained including Lp(a). Neither baseline Lp(a) levels nor relative risk of developing future PAD were found to be statistically significant [65]. Another prospective cohort study which evaluated female health professionals > 45 years old found borderline non-significant results for the association of Lp(a) levels and PAD, defined as intermittent claudication and/or peripheral artery surgery (HR 1.6, 95% CI 1.0–2.6; *p* = 0.07). The lack of statistical significance likely related to a low event rate as only 100 of the 27,935 participants met the primary endpoint of diagnosed PAD [66]. An additional prospective study followed 1222 patients with either intermittent claudication or critical limb ischemia at baseline. They stratified these patients into two cohorts, using two different measurements of serum Lp(a) levels (mg/dL and nmol/L). Lp(a) levels were measured immediately before endovascular repair, and they reported data on cardiovascular mortality during the coming 4.3 and 7.6 years. Despite long follow-ups, they did not find association between Lp(a) levels and cardiovascular death. The endpoint of cardiovascular death was retrospectively collected from the federal death registry. This study had several limitations, including being performed at a single center, measuring Lp(a) in an acute phase, and measuring cardiovascular death as the outcome. It is possible that the investigators missed out on non-fatal cardiovascular events [67].

## 8. Management

While identifying individuals with elevated Lp(a) levels is a problem in and of itself, the next challenge is how to properly manage this high-risk population. There are very few data on the management of PAD in the setting of elevated Lp(a) levels itself. The AHA/ACC 2016 guidelines have a Class 1A recommendation to initiate HMG-CoA reductase (statin) therapy in all patients diagnosed with PAD [68]. Multiple studies have shown that in patients suffering from PAD, initiation of statin therapy improves MACE and MALE [69,70,71,72]. Although no large studies have tested the effects of statin on those suffering from both elevated Lp(a) levels and PAD, the data around statins as a therapeutic option for elevated Lp(a) levels indicate it would likely have little to no effect. While most recent studies indicate that statins do not lower Lp(a) levels [73,74], some suggest that they might actually result in an increase in Lp(a) levels [75,76,77]. Although one could argue that the use of a statin is warranted even in the absence of hyperlipidemia as it could mitigate the downstream effects of atherosclerosis such as plaque stabilization, some data suggest worse cardiovascular events in patients with elevated Lp(a) levels on statins compared to elevated Lp(a) levels on placebo [78]. Based on current data, it does not appear that statins have a beneficial role in the management of elevated Lp(a) levels in the setting of PAD; however, this has yet to be studied.

Protein convertase subtilisin-kexin type 9 (PCSK9) inhibitors have demonstrated the ability to significantly reduce serum Lp(a) levels. Although commercially available, PCSK9i have not been approved by the United States Food and Drug Administration specifically for the management of elevated Lp(a) levels. The proposed mechanisms by which this occurs are increased LDL receptor expression leading to increased clearance of Lp(a) and possibly direct impact on apoB-100 protein and microsomal triglyceride transfer protein activity [79]. The ODYSSEY OUTCOMES Trial was a multicenter, randomized, double-blind, placebo-controlled trial involving 18,924 patients who had acute coronary syndrome within the preceding year who also had LDL cholesterol of at least 70 mg/dL, a non-HDL cholesterol of at least 100 mg/dL, or an apoB level of at least 80 mg/dL, and were receiving statin therapy at high-intensity or maximum tolerated dose. Patients were randomly assigned to receive alirocumab subcutaneously every two weeks or a placebo. They were able to prove that in patients who had previous acute coronary syndrome and were receiving high-intensity statin therapy, recurrent ischemic cardiovascular results were lower in the individuals taking alirocumab compared to placebo; however, no data were reported on PAD outcomes [80]. In a prespecified analysis of the trial, investigators studied the role of lowering serum Lp(a) levels with alirocumab and its effect on PAD events including critical limb ischemia, limb revascularization, and amputation for ischemia. They found that alirocumab markedly decreased the number of PAD events compared to placebo (HR 0.69, 95% CI 0.54–0.89, *p* = 0.004). Additionally, they showed greater absolute and relative risk reductions in patients with higher baseline Lp(a) levels. Lastly, they found a decrease in median serum Lp(a) levels by 23.5% [81]. The FOURIER Trial was a multicenter, randomized, double-blind, placebo-controlled trial involving 27,564 patients with known ASCVD who had LDL ≥ 70 mg/dL and were on statin therapy. They found that evolocumab on the background of statin therapy significantly reduced the risk of cardiovascular events [82]. A subsequent sub-analysis was able to prove that in patients with PAD, evolocumab reduced the risk of both MACE and MALE [83]. Another sub-analysis showed that evolocumab was able to decrease serum Lp(a) levels by a median of 26.9% [84]. Unlike subsequent analyses of the ODYSSEY OUTCOMES, there was no analysis of Lp(a) in PAD using the FOURIER data. The only adverse event which was statistically significant from placebo in both alirocumab and evolocumab was injection sight reaction which was seen at a rate of 3.8% and 2.1%, respectively [80,82].

Although niacin is well known to raise HDL while lowering LDL and triglycerides, it has also been shown to lower serum Lp(a) levels. Although the mechanism by which this occurs is still unclear, it has been proposed that niacin decreases apo(a) transcription and impairs apoB-100 secretion by inhibiting triglyceride synthesis [85,86]. In the Atherothrombosis Intervention in Metabolic Syndrome with Low HDL/High Triglyceride and Impact on Global Health Outcomes (AIM-HIGH) Trial, patients with known cardiovascular disease with low HDL < 40 mg/dL for men and <50 mg/dL for women, elevated triglycerides 150–400 mg/dL, and LDL < 180 mg/dL were randomized to receive either simvastatin plus placebo or simvastatin, plus extended-release niacin 1500 to 2000 mg/day. Cardiovascular disease was defined as stable CAD, CVD, carotid disease, or PAD. They found that there was no benefit of addition of niacin to statin therapy despite improvement in HDL and triglycerides [87]. A subsequent analysis of the AIM-HIGH study found that the niacin group had reduced Lp(a) levels by 21% (80.2 vs. 63.4 nmol/L) compared to baseline; however, this was not associated with any reduction in cardiovascular events [88]. The Heart Protection Study 2–Treatment of HDL to Reduce the Incidence of Vascular Events (HPS2-THRIVE) studied 25,673 patients 50 to 80 years of age who had MI, CeVD, PAD, or diabetes mellitus with evidence of symptomatic CAD. Patients were given randomized niacin in combination with laropiprant or placebo, on top of simvastatin with or without ezetimibe background therapy. They found that the addition of niacin and laropiprant to simvastatin did not reduce major vascular events. Lp(a) levels were measured in 1999 randomly selected patients one year into the study; however, they found that patients taking niacin and laropiprant had levels 17.8% lower than statin alone (50.7 vs. 60.3 nmol/L) [89]. Although both of these studies were able to show that niacin can significantly reduce serum Lp(a) levels, they did not find a change in cardiovascular outcomes. It should be noted that neither study was prospectively designed for patients with elevated Lp(a) levels, and one could argue that the serum Lp(a) levels in the patients studied did not warrant Lp(a) lowering therapy.

Lipoprotein apheresis is a process in which Lp(a) is removed from circulation via filtration, precipitation, or adsorption. Although a 50–75% decrease in serum Lp(a) levels can be seen, these levels may return to baseline in 8–13 days, requiring patients to undergo weekly or biweekly apheresis. Rarely used in most countries, apheresis is mostly seen in Germany where it is used in patients with Lp(a) is >60 mg/dL with progressive atherosclerotic disease despite optimization of other risk factors [90]. A study representing ~60% of patients who were receiving lipoprotein apheresis in Germany investigated 170 patients with elevated Lp(a) levels two years before and after receiving apheresis. They found that compared to the two years prior to lipoprotein apheresis, in the two years following the initiation of lipoprotein apheresis, the rate of peripheral arterial events including angioplasty, stenting, or bypass reduced from 30 to 11 [91]. In a small study on 10 patients with isolated elevations in Lp(a) levels who all had PAD and had recently undergone revascularization procedures, they found ankle-brachial-index increased from 0.5 ± 0.2 to 0.9 ± 0.1 (*p* < 0.001). Additionally, the frequency of revascularization procedures was markedly reduced from 35 revascularization within the 12 months prior to initiating apheresis versus 1 revascularization in the 12 months after initiating apheresis [92].

Cholesterol Transfer Protein (CETP) inhibitors mediate the transfer of cholesterol esters from HDL to apoB-100 containing lipoproteins. Four CETP inhibitors have undergone clinical trials and have shown reduction in Lp(a) levels by as much as 36%; however, they were shown to have only mild clinical benefit [93]. A newer CETP inhibitor, obicetrapib has been shown to reduce Lp(a) levels by as much as 56.5% [94]. Phase 3 investigations are currently underway to assess the safety and clinical benefits, including cardiovascular outcomes. There are currently no CETP inhibitors available on the market.

Lomitapide is a microsomal triglyceride transfer protein inhibitor that has been shown to reduce Lp(a) levels by 15–19%. Microsomal triglyceride transfer protein is responsible for transferring triglycerides to apoB in hepatocytes and enterocytes; therefore, inhibition leads to decrease in apoB-containing molecules such as Lp(a). Currently, Lomitapide is only used in patients with homozygous familial hypercholesterolemia [93,95].

There are two RNA-based therapies in clinical development, siRNAs and antisense oligonucleotides (ASOs). Pelacarsen is an ASO that binds to a complementary RNA sequence, triggering the degradation of apo(a) mRNA and consequently lowering Lp(a) levels. In a phase 2 clinical trial involving 286 patients with established ACVD and Lpa levels > 60 mg/dL, various dosages of pelacarsen (20, 40, or 60 mg every four weeks; 20 mg every two weeks; or 20 mg weekly) were administered. The trial revealed that there was a dose-dependent reduction in Lp(a) levels, with an 80% decrease observed in the group receiving 20 mg pelacarsen weekly [96]. Currently, pelacarsen is undergoing a phase 3 trial to investigate whether the reduction in Lp(a) levels translates into clinical benefit. Similar to ASOs, siRNAs are double-stranded RNA molecules that separate within the cell. The antisense strand then binds to its complementary mRNA target, initiating degradation. The antisense strand is both stable and recyclable, allowing for less frequent dosing. In a randomized, double-blind, placebo-controlled phase 2 trial comprising 281 patients with known ASCVD and a median Lp(a) concentration of 67.5 mg/dL, olpasiran was found to significantly reduce Lp(a) levels in a dose-dependent manner. At higher doses, olpasiran can reduce serum Lp(a) levels by over 95% [97]. There is currently a phase 3 trial investigating the role of olpasiran in reducing CAD, MI, and urgent coronary catheterization in patients with Lp(a) > 200 nmol/L. Other RNA targeting therapeutics being investigated to lower Lp(a) levels include zerlasiran and lepodisiran. Although both have been shown to reduce serum Lp(a) levels by up to 99% and 97%, respectively, their role in improving ASCVD outcomes has yet to be studied [98,99]. There are not currently any data regarding the use of RNA-based therapies in patients with both elevated Lp(a) levels and PAD.

Muvalapin is an oral small molecular inhibitor of Lp(a) that works by binding to kringle IV types 7 and 8, thereby preventing apo(a) from binding to apoB-100. In a phase 1 clinical trial, it has recently been shown to reduce serum Lp(a) levels by as much as 65% in healthy participants with Lp(a) ≥ 30 mg/dL [100]. It has not yet been studied if this is associated with improved ASCVD outcomes.

## 9. Conclusions

Although Lp(a) has been classically associated with CAD, CVD, and calcific aortic stenosis, there is a growing body of evidence that implicates Lp(a) as a causal risk factor for PAD as well. Despite this association, current clinical guidelines provide few recommendations in the management of elevated Lp(a) levels in the setting of any of these conditions. Historically, niacin was a therapeutic mainstay as it has been proven to lower serum Lp(a) levels. However, more recent studies have discounted its role as an effective treatment modality as its use has not been associated with improvement in clinical outcomes. Currently, PCSK9 inhibitors remain the only therapeutic option on the market that has been proven to not only lower serum Lp(a) levels but also reduce adverse outcomes associated with PAD. In light of current and emerging data, there has been great interest in developing further treatment modalities that not only empirically lower serum Lp(a) levels but also have proven clinical efficacy in reducing PAD outcomes. As these new therapeutics become commercially available, clinical trials assessing PAD outcomes will need to be conducted to provide direction in the development of clinical guidelines.

## Figures and Tables

**Figure 1 biomedicines-12-01229-f001:**
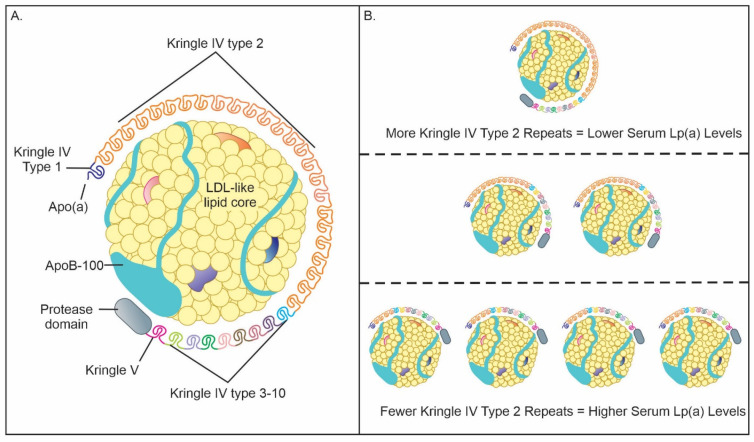
(**A**) Lipoprotein(a) [Lp(a)] structure composed of a low-density-lipoprotein (LDL)-like lipid core containing apolipoprotein B-100 (apoB-100) covalently bound to apolipoprotein(a) [apo(a)] by disulfide bonds. Apo(a) contains single copies of kringle V, kringle IV type 1, kringle IV types 3–10, and a variable number of kringle IV type 2 repeats. (**B**) Serum Lp(a) levels are inversely related to the number of kringle IV type 2 repeats on the apo(a) subunit.

**Figure 2 biomedicines-12-01229-f002:**
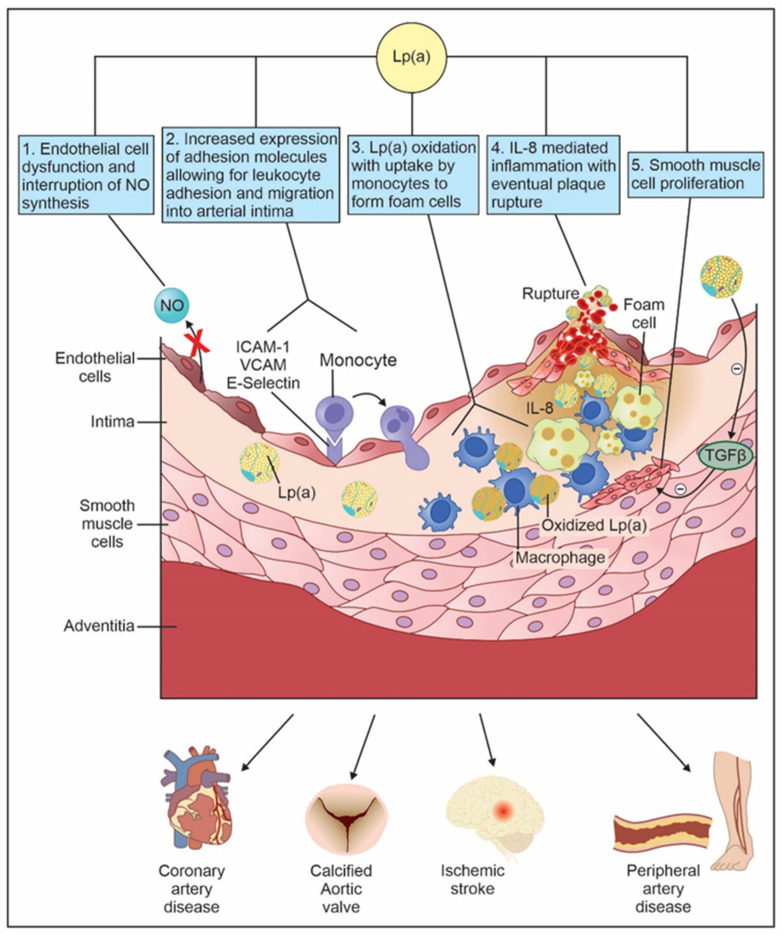
Lp(a) mediated atherogenesis and downstream cardiovascular effects. 1. Lp(a) inhibits nitric oxide synthesis, leading to endothelial cell dysfunction. 2. Lp(a) increases expression of adhesion molecules, allowing for leukocytes to migrate into the arterial intima. 3. Once inside the arterial intima, Lp(a) becomes oxidized and eventually is taken up by monocytes to form foam cells. Once foam cells become overwhelmed by their ability to manage extra lipoproteins, they undergo apoptotic cascade, releasing free radicals, cellular proteases, and inflammatory cytokines. 4. Lp(a) mediates the release of interleukin-8 by macrophages. Interleukin-8 is a key inflammatory cytokine that has chemotactic properties attracting various leukocytes. Interleukin-8 also decreases macrophage expression of metalloproteinases which serve to cleave apo(a), leading to inflammation and plaque rupture. 5. Lp(a) promotes vascular smooth muscle proliferation by indirectly inhibiting transforming factor beta, an inhibitor of smooth muscle cell proliferation. NO: nitric oxide; Lp(a): Lipoprotein(a); ICAM-1: intercellular adhesion molecule-1; VCAM: vascular cell adhesion molecule; IL-8: interleukin-8; TGFβ: Transforming Growth Factor Beta.

**Figure 3 biomedicines-12-01229-f003:**
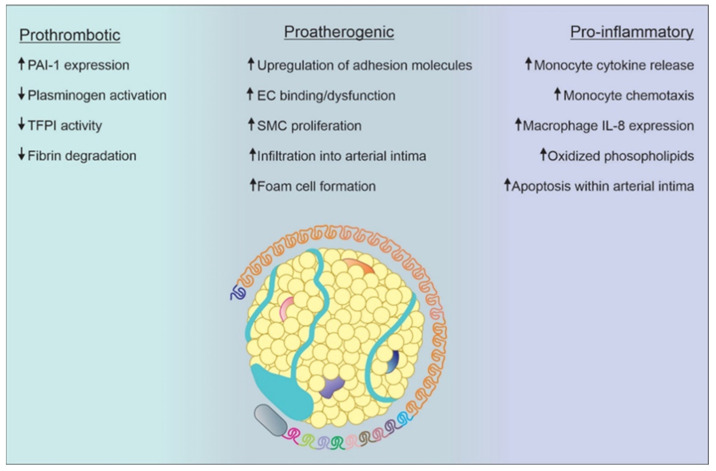
The pathogenicity of Lp(a) is multifactorial and can be classified in three categories: prothrombotic, proatherogenic, and pro-inflammatory. Each of them has its own individual mechanisms. PAI-1: plasminogen activator inhibitor-1; TFPI: tissue factor pathway inhibitor; EC: endothelial cell; SMC: smooth muscle cell; IL-8: interleukin-8.

## Data Availability

Not applicable.

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
