# Peer review of "The Role of Lipoprotein(a) in Peripheral Artery Disease"

_biomedicines, 2024, doi:10.3390/biomedicines12061229_

Round 1

Reviewer 1 Report

Comments and Suggestions for Authors

1.       The introduction can be expanded to provide more context on the clinical significance of peripheral artery disease (PAD) and its relationship to lipoprotein(a) (Lp(a))?

2.       The epidemiological section could delve deeper into the prevalence and distribution of elevated Lp(a) levels in individuals with PAD?

3.       Are there any recent studies or meta-analyses that could be included to enhance the understanding of Lp(a) in PAD?

4.       Are there recent genetic studies or advancements in structural analysis of Lp(a) that could be integrated into this section?

5.       Could any emerging hypotheses or mechanisms regarding Lp(a) in PAD be discussed?

6.       Could the diagnostic testing and screening sections provide more practical guidance for clinicians regarding when and how to assess Lp(a) levels in patients with PAD?

7.       The management section briefly mentions PCSK9 inhibitors but could benefit from further discussion on these agents' efficacy, safety, and availability in treating elevated Lp(a) levels in PAD patients.

8.       The ongoing clinical trials or promising therapeutic approaches targeting Lp(a) in PAD that could be discussed?

9.       The conclusion should summarize key takeaways from the review and outline future research directions, particularly in developing effective therapies for PAD patients with elevated Lp(a) levels?

10.   Do the limitations or gaps in knowledge highlighted in the review warrant further investigation?

Reviewer 2 Report

Comments and Suggestions for Authors

The review article is well-written and provides a comprehensive overview of the current understanding of Lp(a) in relation to peripheral artery disease. It effectively synthesizes epidemiological, genetic, and pathophysiological data to illuminate the complex role of Lp(a) in cardiovascular health. The article highlights the need for increased awareness and better utilization of Lp(a) testing in the clinical setting, particularly for patients with or at risk of PAD. The article also emphasizes the need for further research on effective treatments and the formulation of clinical guidelines to effectively manage elevated Lp(a) levels in these patients. 

1.         Main Question Addressed by the Research:

The review article provides a comprehensive overview of the current understanding of Lp(a) in relation to peripheral artery disease. It effectively synthesizes epidemiological, genetic, and pathophysiological data to illuminate the complex role of Lp(a) in cardiovascular health. The article highlights the need for increased awareness and better utilization of Lp(a) testing in the clinical setting, particularly for patients with or at risk of PAD. The article also emphasizes the need for further research on effective treatments and the formulation of clinical guidelines to effectively manage elevated Lp(a) levels in these patients.

2.         Originality and Relevance:

The authors should include comparisons with coronary artery disease and cerebrovascular disease, as these areas are more established in Lp(a)research, providing a crucial context for understanding its role in PAD.

3.         Contribution to the Subject Area:

This article has the potential to enrich the existing literature by connecting the pathogenic mechanisms of Lp(a), with specific clinical outcomes in PAD. It is important for the review to reference and discuss recent studies to demonstrate how this work either introduces new insights or integrates current understanding. Key studies that should be cited include articles with PubMed IDs 31656821, 37201230, and 33236085, as these are pertinent to the topic and will provide a robust context for the discussion.

4.         Methodological Improvements:

NA

5.         Consistency of Conclusions:

NA

6.         Appropriateness of References:

Key studies on Lp(a)and PAD should be included and appropriately discussed.

7.         Comments on Tables, Figures, and Data Quality:

The tables and figures in the review article are well-presented and easy to understand.

Round 2

Reviewer 1 Report

Comments and Suggestions for Authors

The authors responded to my comments. Now, the manuscript could be accepted for publication.